# CRISPR/Cas9-Mediated Mutagenesis of *Abdominal-A* and *Ultrabithorax* in the Asian Corn Borer, *Ostrinia furnacalis*

**DOI:** 10.3390/insects13040384

**Published:** 2022-04-13

**Authors:** Honglun Bi, Austin Merchant, Junwen Gu, Xiaowei Li, Xuguo Zhou, Qi Zhang

**Affiliations:** 1College of Plant Protection, Shenyang Agricultural University, Shenyang 110866, China; honglunbi@henu.edu.cn (H.B.); gujunwen1995@hotmail.com (J.G.); 2State Key Laboratory of Cotton Biology, Key Laboratory of Plant Stress Biology, School of Life Sciences, Henan University, Kaifeng 475004, China; 3Department of Entomology, University of Kentucky, Lexington, KY 40546, USA; ajme232@g.uky.edu (A.M.); xuguozhou@uky.edu (X.Z.); 4Key Laboratory of Insect Developmental and Evolutionary Biology, Institute of Plant Physiology and Ecology, Shanghai Institutes for Biological Sciences, Chinese Academy of Sciences, Shanghai 200032, China; lixiaoweiyk@163.com

**Keywords:** *Ostrinia furnacalis*, Hox, CRISPR/Cas9, *Abd-A*, *Ubx*

## Abstract

**Simple Summary:**

Homeotic genes encode transcription factors that coordinated the anatomical structure formation during the early embryonic development of organisms. In this study, we functionally characterized two homeotic genes, *Abdominal-A* (*Abd-A*) and *Ultrabithorax* (*Ubx*), in the Asian corn borer, *Ostrinia furnacalis* (a maize pest that has devastated the Asia-Pacific region) by using a CRISPR/Cas9 genome editing system. Our results show that the mutagenesis of *OfAbd-A* and *OfUbx* led to severe morphological defects in *O. furnacalis*, which included fused segments and segmental twist during the larval stage, and hollowed and incision-like segments during the pupal stage in *OfAbd-A* mutants, as well as defects in the wing-pad development in pupal and adult *OfUbx* mutants. Overall, knocking out *Abd-A* and *Ubx* in *O. furnacalis* resulted in the embryonic lethality to, and pleiotropic impact on, other homeotic genes. This study not only confirms the conserved body planning functions in *OfAbd-A* and *OfUbx*, but it also strengthens the control implications of these homeotic genes for lepidopteran pests.

**Abstract:**

(1) Background: *Abdominal-A* (*Abd-A*) and *Ultrabithorax* (*Ubx*) are homeotic genes that determine the identity and morphology of the thorax and abdomen in insects. The Asian corn borer, *Ostrinia furnacalis* (Guenée) (Lepidoptera: Pyralidae), is a devastating maize pest throughout Asia, the Western Pacific, and Australia. Building on previous knowledge, we hypothesized that the knockout of *Abd-A* and *Ubx* would disrupt the abdominal body planning in *O. furnacalis*. (2) Methods: CRISPR/Cas9-targeted mutagenesis was employed to decipher the functions of these homeotic genes. (3) Results: Knockout insects demonstrated classical homeotic transformations. Specifically, the mutagenesis of *OfAbd-A* resulted in: (1) Fused segments and segmental twist during the larval stage; (2) Embryonic lethality; and (3) The pleiotropic upregulation of other homeotic genes, including *Lab*, *Pd*, *Dfd*, *Antp*, and *Abd-B*. The mutagenesis of *OfUbx* led to: (1) Severe defects in the wing pads, which limited the ability of the adults to fly and mate; (2) Female sterility; and (3) The pleiotropic upregulation of other homeotic genes, including *Dfd*, *Abd-B*, and *Wnt1*. (4) Conclusions: These combined results not only support our hypothesis, but they also strengthen the potential of using homeotic genes as molecular targets for the genetic control of this global insect pest.

## 1. Introduction

Asian corn borer, *Ostrinia furnacalis* (Guenée) (Lepidoptera: Pyralidae), is a devastating maize pest throughout Asia, Western Pacific, and Australia [1,2]. Damage caused by *O. furnacalis* results in a loss of approximately 9 million tons of corn annually [3]. Traditional pest management strategies for *O. furnacalis* and European corn borer, *Ostrinia nubilalis* (Hübner), rely on chemical pesticides; biocontrol agents, such as the parasitoid wasp, *Trichogramma ostriniae* [4,5,6,7]; and biopesticides, such as Bt toxins [8,9,10,11]. Bt resistance, however, has developed in *O. furnacalis* in the laboratory [12], and in *O. nubilalis* in the field [13,14,15]. Multiple strategies have been developed to manage Bt resistance, including refuge, stacked traits, and gene editing techniques [16].

Hox genes, which are structurally conserved with identical Homeodomain motifs across taxa [17], dictate the anterior–posterior body identity and morphological features in insects [18,19]. Genes in the Hox family are subcategorized into two clusters: the Antennapedia complex (ANT-C) and the Bithorax complex (BX-C) [20]. There are a total of 8 Hox genes: Labial (lab), *Antennapedia* (*Antp*), *proboscipedia* (*pb*), *Deformed* (*Dfd*), and *Sex combs reduced* (*Scr*) belong to the ANT-C cluster, while *Ultrabithorax* (*Ubx*), *Abdominal-A* (*Abd-A*), and *Abdominal-B* (*Abd-B*) belong to the BX-C cluster [21]. Hox genes are typically expressed at embryonic stage to specify the arrangement of thorax and abdominal segments [17]. Mechanisms that underlie how Hox genes work, individually or collaboratively, are still not fully understood [22,23].

*Abd-A* is essential in the formation of embryonic and abdominal segments in insects [24]. In *Drosophila*, *Abd-A* plays integral roles in suppressing limb development and in specifying abdominal segments [25,26]. Mutation of *Abd-A* results in homeotic transformation of parasegments 5 and 6 during the embryonic stage, which eventually leads to neonatal mortality and defects in head and thorax [17]. *Abd-A* is also involved in the formation of female internal genitalia [27,28], nervous system, fat body [29], and midgut [30]; aorta and heart cardioblast differentiation [31]; and ectopic pigmentation [32].

*Ubx* acts as a genetic switch to modify specific morphological features in thoracic region in insects [17,33,34,35]. In *Drosophila*, *Ubx* promotes the formation of halteres through expression in the meso and metathorax [36,37]. The size and identity of the appendages is modulated by the expression of *Ubx* in larvae [33,35,36,37,38,39,40,41]. In brown planthopper, *Nilaparvata lugens* (Stål), *Ubx* is the key regulator of the transformation between short and long wing forms [22]. In lepidopterans, *Ubx* is expressed in the metathoracic segment in *Bombyx mori* and it regulates wing development [38]. Silencing of *Ubx* and *Abd-A* leads to suspended limb development in *Drosophila*, and in many lepidopteran insects, through the suppression of *Distal-less* (*Dll*) gene expression [27,39,40,41].

The overall goal of this study was to gain a better understanding of body planning and organ development in *O. furnacalis*. Building on previous knowledge and preliminary research, we hypothesized that *Abd-A* and *Ubx* are involved in body patterning of thoracic and abdominal segments in *O. furnacalis*. To examine this overarching hypothesis, we: (1) Established a CRISPR/Cas9 genome editing system in *O. furnacalis*; and (2) Functionally characterized *Abd-A* and *Ubx* in this devastating corn pest.

## 2. Materials and Methods

### 2.1. Phylogenetic Analysis of OfAbd-A and OfUbx Genes

#### 2.1.1. Phylogenetic Tree

Abd-A protein sequences that were used in the phylogenetic analysis included *Apis mellifera* (GenBank accession number: XP_016772643.1), *Aedes aegypti* (NP_001345961.1), *Tribolium castaneum* (NP_001034518.1), *Bombyx mori* (NP_001166808.1), *Drosophila melanogaster* (NP_001247145.1), *Danaus plexippus plexippus* (OWR52832.1), *Pieris rapae* (XP_022113611.1), *Bicyclus anynana* (XP_023954095.1), *Vanessa tameamea* (XP_026485495.1), *Papilio xuthus* (XP_013173869.1), *Dendrolimus punctatus* (AQM32554.1), *Plutella xylostella* (XP_011569267.1), *Spodoptera litura* (XP_022825998.1), *Trichoplusia ni* (XP_026731481.1), *Myrmica rubra* (AAK06846.2), *Papilio polytes* (XP_013139462.1), and *Galleria mellonella* (XP_026759670.1). *Ubx* protein sequences included *Drosophila melanogaster* (NP_536752.1), *Bombyx mori* (NP_001107632.1), *Apis mellifera* (NP_001162171.1), *Ceratitis capitata* (XP_004524337.1), *Spodoptera litura* (XP_022837384.1), *Trichoplusia ni* (XP_026735287.1), *Bactrocera dorsalis* (XP_011203163.1), *Plutella xylostella* (NP_001303599.1), *Tribolium castaneum* (NP_001034497.1), *Anopheles gambiae* (AAC31942.1), *Sogatella furcifera* (ATW63192.1), *Papilio xuthus* (XP_013173873.1), *Pieris rapae* (XP_022113620.1), *Helicoverpa armigera* (XP_021195138.1), *Dendrolimus punctatus* (AQM32553.1), *Orchesella cincta* (CDI44538.1), *Papilio machaon* (XP_014359749.1), *Bicyclus anynana* (XP_023954092.1), *Biston betularia* (ADO33070.2), and *Galleria mellonella* (XP_026759674.1). Sequence alignment was constructed by using the maximum likelihood method based on CLUSTAL W2 [42,43]. All of the ambiguous positions were removed for each sequence pair. The neighbor-joining method was used to create the tree from 18 available *Abd-A* sequences and 21 *Ubx* sequences. The percentages of replicate trees in which the associated taxa clustered together in the bootstrap test (1000 replicates) are shown next to the branches. The evolutionary distances were computed using the Poisson correction method and are displayed as the number of amino acid substitutions per site. The evolutionary history was inferred by using the neighbor-joining method and MEGA-7 with 1000 bootstrap replicates [44].

#### 2.1.2. Multiple Alignment

Amino acid sequences encoded by *Abd-A* from *Drosophila melanogaster* (NP_001247145), *Plutella xylostella* (XP_011569267), *Bombyx mori* (ACD10794), *Tribolium castaneum* (NP_001034518), and *Apis mellifera* (XP_016772643), as well as the predicted amino acid sequence of *Abd-A* from *Ostrinia furnacalis*, and similarly, amino acid sequences of *Ubx* derived from *Drosophila melanogaster* (NP_536752), *Plutella xylostella* (NP_001303599), *Bombyx mori* (NP_001107632), *Tribolium castaneum* (NP_001034497), *Apis mellifera* (NP_001162171), and *O. furnacalis* were analyzed by Clustal Omega, which is a multiple sequence alignment program that uses seeded guide trees and HMM profile–profile techniques to generate alignments between three or more sequences [45].

### 2.2. Temporal and Spatial Expression Profiles of OfAbd-A and OfUbx

#### 2.2.1. Sample Collection

To investigate the spatial expressions of *OfAbd-A* and *OfUbx* genes, total RNA was isolated from eggs, from larvae during the first day of each instar, from prepupae (PP), from pupae (P), and from adults (A), by using Trizol reagent (Invitrogen, Carlsbad, CA, USA), and it was treated with RNase-free DNase I (Ambion, Austin, TX, USA), according to the manufacturer’s protocols. For tissue-specific analysis, fifth-instar larvae were dissected three days after ecdysis to obtain tissue from head, midgut, foregut, fat body, epidermis, testes, and ovaries, which were kept in liquid nitrogen for sample collection and total RNA extraction. To address the question of how Hox proteins are regulated by the mutagenesis of *OfAbd-A* and *OfUbx*, we collected pupal *O. furnacalis* for the total RNA extraction to analyze the relative transcript levels of the genes in the Hox family in *OfAbd-A* and *OfUbx* mutants.

#### 2.2.2. Reverse Transcription Quantitative Real-Time PCR (RT-qPCR) Analysis

cDNAs were synthesized using 1 μg total RNA from each developmental stage as templates by using an Omniscript reverse transcriptase kit (Qiagen, Hilden, Germany) in a 20 μL reaction mixture. The primers used for the RT-qPCR are listed in Appendix A. RT-qPCR was performed on an Eppendorf Real-time PCR System using the following conditions: a 2 min denaturing cycle at 95 °C, 35 cycles of 1 min at 95 °C, 30 s at 55 °C, and 30 s at 72 °C, followed by a final extension at 72 °C for 10 min.

### 2.3. CRISPR/Cas9-Targeted Mutagenesis in O. furnacalis

#### 2.3.1. Molecular Cloning and Target Selection

Total RNA was isolated from fifth-instar larvae using Trizol Reagent (Invitrogen, Carlsbad, USA) and was treated with RNase-free DNase I (Ambion, Austin, TX, USA), according to the manufacturer’s protocol. cDNA was synthesized with Omniscript reverse transcriptase kit (Qiagen, Hilden, Germany), using manufacturer’s instructions. Putative *OfAbd-A* and *OfUbx* genes were identified using the NCBI Blast system. *OfAbd-A* cDNA fragments were amplified by PCR with the following pair of primers: forward, 5′-ATGGCAGCGGCTGCCCAGTT-3′; and reverse, 5′-TTACGTGGGGACTTTGTTCA-3′. *OfUbx* cDNA fragments were amplified by PCR with the following pair of primers: forward, 5′-ATGAACTCCTACTTTGAGCAGGGT-3′; and reverse, 5′-TTACGTGGGGACTTTGTTCA -3′. PCR was carried out using KOD -Plus- polymerase (TOYOBO, Osaka, Japan) under the following conditions: 98 °C for 2 min, followed by 30 cycles at 98 °C for 30 s, 55 °C for 30 s, 68 °C for 1 min, and an elongation phase at 68 °C for 10 min. Amplified products were sequenced after cloning into a PJET1.2-T vector (Fermentas, Burlington, ON, Canada). The primers are listed in Appendix A.

#### 2.3.2. Synthesis of Cas9 mRNA and sgRNAs

We selected one 23 bp sgRNA to target the *OfAbd-A* genome locus on the third exon of the common region in the four spliced variants, and two 23 bp sgRNAs targeting the *OfUbx* genome locus in exon 1 of the common region for the two spliced variants. The sgRNA was subcloned into a 500 bp linearized CloneJet PJET1.2-T vector (Thermo Fisher, Waltham, MA, USA), upstream of the protospacer adjacent motif (PAM) sequence to allow sgRNA expression under the control of the T7 promoter. The sgRNA was synthesized in vitro with a MEGAScript T7 kit (Ambion, Austin, TX, USA), according to the manufacturer’s instructions. Cas9 mRNA was synthesized in vitro using the mMESSAGE T7 Kit (Ambion, Austin, TX, USA) and a PTD1-T7-Cas9 vector as the template, according to the manufacturer’s instructions.

#### 2.3.3. Embryo Microinjection

A laboratory strain (Shanghai) of *O. furnacalis* larvae was reared with an artificial diet [46]. Insects were kept at 25 °C with 80% relative humidity and a 16:8 light:dark photoperiod [47]. Adults of *O. furnacalis* were maintained in transparent plastic bags and were fed using cotton balls soaked in sugar water (10% honey in distilled water) to allow them to lay eggs. Eggs were collected from the plastic bags, which were cut into pieces for easier collection, and were arranged in a row on sterilized cover slips, as described previously [48]. Eggs were injected on the lateral side with 10 nL of a mixture containing 300 ng/μL Cas9 mRNA and 150 ng/μL sgRNA, within 1 h of oviposition. After injection, eggs were incubated in a humidified chamber at 25 °C for 4 days until hatching.

### 2.4. Genomic DNA Extraction to Identify Successful Mutants

The genomic DNA of *O. furnacalis* larvae was extracted by using phenol:chloroform and an isopropanol precipitation extraction. Specifically, newly hatched larvae were collected and incubated with proteinase K, followed by DNA purification and RNaseA treatment. PCR was carried out to identify *OfAbd-A* and *OfUbx* mutant alleles by using primers spanning the target sites in *OfAbd-A* and *OfUbx* (Appendix A). The PCR conditions were as follows: 98 °C for 2 min, followed by 35 cycles of 94 °C for 10 s, 55 °C for 30 s, and 72 °C for 1 min, followed by a final extension period of 72 °C for 10 min. The PCR products were cloned into pJET1.2-T vectors (Fermentas, Burlington, ON, Canada) and were sent for sequencing. The mutants were photographed with a digital stereoscope (Nikon AZ100).

To detect *OfAbd-A* and *OfUbx* mutants, RT-qPCR was carried out using gene-specific primers for *Abd-A* (forward, 5′-CGGCAAACTTACACGAGGTT-3′; and reverse, 5′-TCCTGCTCCTCTCTCTCTCG-3′) to amplify a 221 bp fragment. RT-qPCR reactions were carried out by using gene-specific primers for the *Ubx* gene (forward, 5′-CCACACCTTCTACCCTTGGA-3′; and reverse, 5′- TCATCCTCCGATTCTGGAAC-3′) to amplify a 221 bp fragment. *Ofactin* was used as an internal control for the RT-qPCR [47]. The primers are listed in Appendix A.

### 2.5. Analysis of Hatch Rate after Mutagenesis

We analyzed the hatching rate of eggs injected with different concentrations of *OfAbd-A* sgRNA, and the hatching rate of eggs produced by adult *OfUbx* mutants. When *OfUbx* mutants reach adult stage, single-paired adults of different combinations, including mutant male with mutant female, mutant female with WT male, mutant male with WT female, and GFP male with GFP female, as control, were collected to examine hatching rate of G1. Each combination was set in five replicates. All adults were placed in a transparent plastic bag supplemented with cotton balls soaked in 10% honey water to allow them to lay eggs. After five days, we collected eggs from each bag and allowed them to hatch in order to analyze hatching rate of each combination.

### 2.6. Statistical Analysis

RT-qPCR results were collected and presented in figures by using GraphPad Prism 7.0 software (GraphPad, San Diego, CA, USA). Data were analyzed using an unpaired Student’s *t* test (SPSS Statistics 25.0 software, IBM, Armonk, NY, USA). Probability values of less than 0.05 were considered significant. Data are presented as means with SEMs.

## 3. Results

### 3.1. Phylogenetic Analysis of OfAbd-A and OfUbx

Hox genes in both Drosophila and other invertebrates perform the same overall function of body organization along the anterior–posterior axis [49]. In Drosophila, bithorax complex (BX-C) includes Ubx, Abd-A, and Abd-B genes, which implies that Ubx and Abd-A have a close relationship [50]. We analyzed the genomic sequence of the *Abd-A* and *Ubx* genes in *O. furnacalis* and performed multiple alignments to carry out a sequence analysis. The genomic sequence of *OfAbd-A* is 41,431 base pairs (bps) in length, and it has three exons with four alternative splicing variants. *OfAbd-A* mRNA contains four ORFs that are 1050, 1050, 1035, and 1023 bp in length, which encode proteins of lengths of 349, 349, 344, and 340 amino acids (aa), respectively. Comparably, length of *Abd-A* in *Drosophila melanogaster* is 990 bp, which encodes a 330aa protein, which exhibits a highly conserved domain with *OfAbd-A*. Among other moth species, open reading frame (ORF) of *Abd-A* in *Bombyx mori* (NP_001166808.1) is 1056 bp, which encodes a protein of 351aa, while, in *Plutella xylostella* (XP_011569267), 1062 bp *Abd-A* encodes a protein of 352aa. Both genes are slightly larger than the *O. furnacalis Abd-A* gene. The genomic sequence of *OfUbx* is 123,000 bp in length and it has two exons, with two ORFs of 774 and 762 bp, which encode 257 and 254aa proteins, respectively. *Ubx* encodes a 389aa protein in *D. melanogaster* (NP_536752.1) and a 254aa protein in *B. mori* (NP_001107632.1). On the basis of our phylogenetic analysis, *OfAbd-A* is clustered with homologs in other Lepidopteran species, such as *Plutella xylostella*, *Bombyx mori*, and *Papilio Xuthus* (Figure 1A), while *OfUbx* is grouped with homologs in *Dendrolimus punctatus* and *Biston betularia* (Figure 1B). Our sequence alignment analysis shows that *OfAbd-A* and *OfUbx* both contain a Homeodomain and the respective Abdominal and Ultrabithorax domains, which demonstrates a structural conservation with their homologues in other insects (Figure 2A,B).

### 3.2. Temporal–Spatial Distribution of OfAbd-A and OfUbx

To investigate the transcript changes of *OfAbd-A* and *OfUbx* during different developmental stages in *Ostrinia furnacalis*, we collected eggs, larval instars (first–fifth), from hatching to wandering stages, pupae, and adults of both females and males for the total RNA extraction and RT-qPCR analysis. The results of the RT-qPCR show that the *OfAbd-A* gene is highly expressed in the egg, wandering, pupal, and female adult stages (Figure 3A), whereas the *OfUbx* gene is highly expressed in the pupal and adult stages of both females and males (Figure 3B).

To address the question of whether the transcript levels of *OfAbd-A* and *OfUbx* are tissue specific, we investigated the expression levels of these two genes from different tissue regions in *O. furnacalis* larvae by using RT-qPCR. Tissue from head, foregut (FG), midgut (MG), fat body (FB), epidermis (EPI), testes in male larvae (TE), and ovaries in female larvae (OV) were dissected using three-day-old fifth-instar larvae (L5D3). Results from RT-qPCR analysis show that *OfAbd-A* is highly expressed in the epidermis and ovaries (Figure 3C), while *OfUbx* is highly expressed in fat body, epidermis, and ovaries (Figure 3D).

### 3.3. CRISPR/Cas9-Mediated Mutagenesis in O. furnacalis

We used CRISPR/Cas9 genome editing system to knock out *Abd-A* and *Ubx* genes in *O. furnacalis* [32]. There are four isoforms of *OfAbd-A*, and two isoforms of *OfUbx* (Figure 4A and Figure 5A). To identify the mutated alleles, we extracted and sequenced the genomic DNA from larvae with mutant phenotypes for *OfAbd-A*, and from pupae with mutant phenotypes for *OfUbx*. The genome sequencing showed the successful deletion of sequences within a single target site in the *OfAbd-A* gene (Figure 4C), and within two target sites in the *OfUbx* gene (Figure 5C).

### 3.4. Phenotypic Impacts of OfAbd-A and OfUbx Mutagenesis

#### 3.4.1. Morphological Impacts

The *OfAbd-A* knockout resulted in twisted abdominal segments in the larvae via the abnormal combination of adjacent segments from A2 to A7 (Figure 6A). In the pupal stage, mutagenesis of *OfAbd-A* led to hollowed and incision-like segments at abdominal segments, A2 to A7 (Figure 6B). *OfUbx* gene mutagenesis caused abnormal folding of wing in pupal stage, which hinders the complete covering of the thoracic region (Figure 7A). Mutagenesis of *OfUbx* also led to the disruption of wing development in pupae. Moreover, severe mutagenesis prevented the adults from spreading their wings to fly, which thus limited their ability to mate and reproduce (Figure 7B).

#### 3.4.2. Physiological Impacts

To investigate the physiological changes induced by *OfAbd-A* mutation in *O. furnacalis*, we analyzed the hatching rate of larvae post *OfAbd-A* injection by using GFP as a control (Table 1). The hatching rate of the GFP control larvae was approximately 62%, while that of the embryos injected with *OfAbd-A* Cas9/sgRNA was approximately 30%, which suggests that the decrease in hatching rate was specifically caused by *OfAbd-A* mutagenesis. To further clarify the relationship between the hatching rate and *OfAbd-A* mutagenesis, *O. furnacalis* eggs were injected with different doses of Cas9 and *Abd-A* sgRNA. Results show that, when eggs were injected with Cas9/sgRNA at a concentration of 300 ng/μL, the hatching rate of the larvae decreased to 21% with a mutation rate of approximately 56% (Table 1). When the injection concentration of Cas9/sgRNA was decreased to 150 ng/μL, the hatching rate of the larvae reached 33%, and the mutation rate decreased to approximately 38%. Pupation rate of the treatment group was also lower than that of the GFP control group (Table 1). Our results demonstrate that mutagenesis of the *OfAbd-A* gene leads to embryonic lethality and affects larval development (Figure 8).

Similarly, higher concentrations of *OfUbx* sgRNA and Cas9 mRNA induced a higher mutation rate. Only 50% of the pupal *OfUbx* mutants were able to emerge to the adult stage (Table 1), compared to the emergence rate of GFP Cas9/sgRNA-treated pupae, which was 71%. We also observed that the adult male *OfUbx* mutants were unable to mate with either wild-type or mutant females, and that female *OfUbx* mutants could not mate with males from either group. On the basis of our observations, *OfUbx* mutation can induce sterility in both sexes of adult *O. furnacalis* (Figure 8).

#### 3.4.3. Pleiotropic Impacts

To address the question of how Hox proteins are regulated by the mutagenesis of *OfAbd-A* and *OfUbx*, we analyzed the relative expression levels of the other Hox genes within the *OfAbd-A* and *OfUbx* mutants in the pupal stage. *OfAbd-A* mutagenesis gave rise to the pleiotropic upregulation of *Labial* (*Lab*), *Proboscipedia* (*Pb*), *Deformed* (*Dfd*), *Antennapedia* (*Antp*), and *Abd-B* (Appendix A), while *OfUbx* mutagenesis resulted in the upregulation of *Dfd*, *Abd-B*, and *Wingless Integrated family member 1* (*Wnt1*) (Appendix A). The relative transcript level of *Dfd* increased dramatically following *OfUbx* and *OfAbd-A* mutagenesis. Our results show no significant transcript variation in *Scr* in response to *OfUbx* and *OfAbd-A* mutagenesis, while *Antp* was highly upregulated after *OfAbd-A* was knocked out (Appendix A).

## 4. Discussion

In insects, homeotic genes, such as *Abd-A* and *Ubx*, specify the distinct identities of body segments [51,52]. We examined the transcription levels of *OfAbd-A* and *OfUbx* across *O. furnacalis* developmental stages to generate a temporal profile of expression for these two genes. *OfAbd-A* is highly expressed in eggs, wandering-stage larvae, pupae, and female adults (Figure 3A). In other insect species, *Abd-A* has been reported to initiate its function of segmental identity determination during embryogenesis [53]. *OfUbx* is highly expressed in pupal and adult stages (Figure 3B). This result is in accordance with characterized function of *Ubx* in rice planthoppers, where *Ubx* is expressed in both forewing and hindwing, and it acts as the key regulator for the switch between long and short wing forms in adult stages of both sexes [22]. In addition, *Ubx* was reported to regulate the development of A1 and T3 legs during adult stage in *Oncopeltus fasciatus* [52].

Spatial expression pattern of Hox family genes is relatively well understood in Drosophila [19,54] and other insects [55,56], especially in the thoracic and abdominal segments. However, studies on tissue-specific characterization in whole larvae are comparably lacking. We investigated the expression of *OfAbd-A* and *OfUbx* in seven larval tissue types in *O. furnacalis* (Figure 3C,D). Our results show that the relative expression of *OfAbd-A* is dramatically higher in epidermis, and is comparably high in ovary tissue, compared to other body regions (Figure 3C), which is in accordance with previous reports that show its expression in the epidermal and neural cells in Drosophila [19,26]. The *OfUbx* gene is highly expressed within epidermis, fat body, and ovary tissues (Figure 3D), which is consistent with previous studies that show that *Ubx* is involved in gut and muscle development in Drosophila and that it has distinct identities in segmental and structural identification [26,57]. The extensive expression of *Ubx* was examined in Drosophila in a contiguous region from T3 to A6/A7 in abdomen and gonads, alongside the expression in the fat body [29]. The expression of *Ubx* was also reported to extend to cover anterior and posterior body regions, and to suppress limb and/or wing development [58,59,60].

CRISPR/Cas9 is a genome editing tool that has recently been exploited as an alternative for pest management [61,62,63]. In this study, we hypothesized that *OfAbd-A* and *OfUbx* genes modulate the morphological identity and development of thorax and abdomen in *O. furnacalis*. The first gene editing system in *O. furnacalis* was established by You et al. (2018) to characterize the functional role of *Ago1* in pigmentation [48], which was then followed by precise gene manipulation targeting *yellow* [46]. In addition to *Ago1* and *yellow*, research regarding the functional role of Bt toxin receptors, such as ABCC2 and cadherin in *O. furnacalis*, has been conducted by using gene editing systems [47,64]. By using the CRISPR/Cas9 genome editing system, we successfully induced deletion mutations of 59 and 250 bp into *OfAbd-A*, respectively (Figure 4), and deletion mutations of 9 and 82 bp into *OfUbx*, respectively (Figure 5), in *O. furnacalis*. After knocking out *OfAbd-A*, the abnormal development of the abdominal segments was observed in larvae (Figure 6), while the abnormal folding of the wings in pupae and adults was observed after *OfUbx* mutagenesis (Figure 7A,B). In the latter case, adults with severely malformed wings were unable to fly, which restricted their ability to mate and reproduce.

*Abd-A* regulates wing disc cuticle protein genes that control larval-to-pupal metamorphosis in *Bombyx mori*, and silencing of *Abd-A* has demonstrated its role in patterning the third to sixth abdominal segments during embryonic development [41]. In *Spodoptera litura*, *Abd-A* is essential for larval segmentation, and mutation can lead to ectopic pigmentation during embryonic development [32]. Phenotypic impacts of *Abd-A* mutants observed in *O. furnacalis* (Figure 6) are consistent with McGinnis et al. (1992), which are that *Abd-A* modulates the development of posterior abdominal segment 1 to anterior abdominal segment 7 in *Drosophila* [19].

*Ubx* carries similar functions for wing development and scale morphology in different insect species [38,65]. It modulates the transformation of wings to halteres in third thoracic segment in Dipterans through the repression of wing-specific genes [65]. *Ubx*, in beetle, Tribolium, regulates the development of elytra in larvae, while RNAi on both *Ubx* and *Abd-A* results in the formation of elytron primordia in all abdominal segments [66]. The short-and-long-wing switch in rice planthoppers is also regulated by *Ubx* gene, and nutritional status of the brown planthopper affects the expression of *Ubx*, thereby influencing the development between short and long wing forms [22]. This observation is in accordance with our result that shows that knockdown of *Ubx* disrupts the wing development in *O. furnacalis* (Figure 7). In our study, knockout of *OfUbx* gene induced disabled mating behavior and decreased the fecundity of adults (Figure 7B). Our research on *OfUbx* mutagenesis verifies the functional role of *Ubx* gene in mediating wing development, from pupal to the adult stages, in *O. furnacalis* (Figure 7A).

Hox genes play important roles in thoracic and abdominal segmentation, but the genetic relationship among genes in the Hox family is still controversial [1]. Hox genes encode functional transcription factors that modulate distinct identities of thoracic and segmental structures through interactions with a series of downstream genes [67]. To further characterize the role of *OfAbd-A* and *OfUbx* in the Hox family, we examined the relative transcript level of Hox genes after mutagenesis (Appendix A). *Abd-A* control the expression of *Lab*, which is localized in gut endothelial cells in the visceral mesoderm [20]. Given that the relative transcript level of *Lab* dramatically increased after *OfAbd-A* mutagenesis in the pupal stage, the role of *Lab* may be to compensate for the lost expression of *OfAbd-A*, and to replicate its function (Appendix A). In addition, *lab* was reported to specify the intercalary and posterior head segments, while *pb*, together with *Scr*, regulates the development of proboscis [68]. *Dfd* and *Scr* act downstream of *Abd-A* to define antennal identity in head segments [69]. *Dfd* is responsible for the development of eye-antennal discs [70]. In our study, increased expression pattern of *Dfd* in the *OfAbd-A* mutants agrees with the report that states that *Dfd* is involved in the homeotic transformation of thoracic epidermis formation, in addition to being responsible for the ventral and labial epidermis formation on head [49]. As *Antp* is involved in the development of thoracic segments in *B. mori* [71], we also examined its relative expression level in *OfAbd-A* and *OfUbx* mutants (Appendix A). Our study shows that *Antp* was highly expressed after *OfAbd-A* mutation, which suggests that it may act as an essential factor in controlling thoracic and abdominal development (Appendix A). *Abd-A* could activate the wingless gene to induce the extra formation of the abdominal segment [72]. In our study, the expression of Abd-B was dramatically increased, while the expression of *Wnt1* was significantly decreased in the *OfAbd-A* mutant (Appendix A), which further confirms the role of *Abd-B*, which suppresses the wingless gene through suppression of *Abd-A* in the A7 region for the elimination of abdomen in Drosophila [72].

Analysis of the downstream gene in *OfUbx* mutant found that *Dfd* was upregulated after *OfUbx* was knocked out (Appendix A), which suggests that *Dfd* could be suppressed by *Ubx*. A previous study on *Drosophila* and *B. mori* shows that *Ubx* regulates a series of downstream genes, such as *Scr* and *wingless* [38]. Our result found that the expression of *Abd-**A* was significantly downregulated, while the expressions of *Abd-B* and Wnt1 were highly elevated in the *OfUbx* mutants (Appendix A).

*Ostrinia furnacalis* is one of the most destructive pests of maize, especially in China and northeast Asia [73]. The extensive use of chemical pesticides has led to adverse environmental and non-target impacts, and the development of resistance has also become a serious concern for *O. furnacalis* management [73]. Therefore, novel pest management approaches, such as RNA-based control alternatives, are urgently needed [74,75].

Our study demonstrates that disruption of *OfAbd-A* can lead to embryonic lethality, and that *OfUbx* mutation induces adult sterility (Figure 8), which suggest their potential as targets of genetic tools for pest management. Since the transgenic line in *O. furnacalis* has been constructed for genetic control by using a piggybac transposon [76], the transgenic insect technique can be carried out by targeting Hox genes, such as *OfAbd-A* and *OfUbx*, in order to exert female-specific interruption (Appendix A). By releasing transgenic adult males that carry a female-specific promoter that initiates the Cas9 protein and the U6 promoter to drive the targeted gene-specific sgRNA expression, males that mate with females in the field can induce female-specific lethality and sterility in the next generation as a form of pest control (Appendix A). Future research regarding how to target *OfAbd-A* and *OfUbx* to induce the abnormal development of pupae and adults in the field is highly warranted. The two Hox genes that are identified and characterized in this study provide fundamental knowledge regarding the development of *O. furnacalis* and show potential as targets for genetic pest control that would be beneficial to maize growers, and that could also extend to other lepidopteran pests (Appendix A).

## 5. Conclusions

In this study, we hypothesized that *OfAbd-A* and *OfUbx* modulate morphological identity and development of thorax and abdomen in *O. furnacalis*. By using a newly developed CRISPR/Cas9 genome editing system in *O. furnacalis*, we knocked out *OfAbd-A* and *OfUbx* respectively, and subsequently observed substantial phenotypic impacts during insect development. Specifically, after *OfAbd-A* mutagenesis, abnormal development of abdominal segments was observed in larvae (Figure 6). The abnormal folding of the wings in pupae and adults was observed after *OfUbx* mutagenesis, which prevented the wings from completely covering the thoracic region of the pupae (Figure 7A), or limited the ability of adults to fly for the purposes of mating or reproducing (Figure 7B). Interestingly, the disruption of *OfAbd-A* led to embryonic lethality, while *OfUbx* mutation induced adult sterility. These combined results not only support our hypothesis, but they also provide a potential molecular target in homeotic genes for the genetic control of this global insect pest.

## Figures and Tables

**Figure 1 insects-13-00384-f001:**
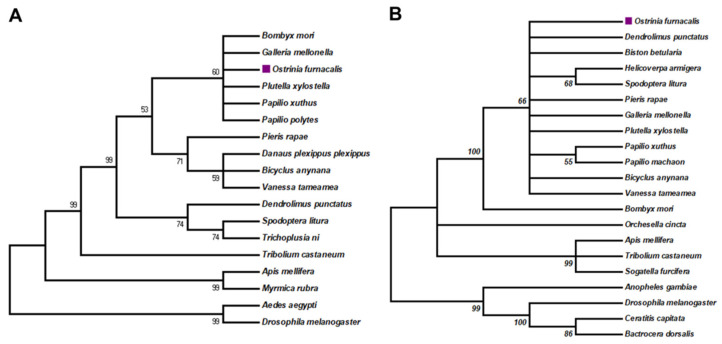
**Phylogenetic analysis of *OfAbd-A* and *OfUbx* genes.** The evolutionary histories of *Abd-A* and *Ubx* were inferred using the neighbor-joining method. The percentages of replicate trees in which the associated taxa clustered together in the bootstrap test (1000 replicates) are shown next to the branches. Evolutionary distances were computed using the Poisson correction method and are listed as the number of amino acid substitutions per site. (**A**) The phylogenetic analysis of *Abd-A* involved 18 amino acid sequences. *OfAbd-A* was denoted by a square. (**B**) The phylogenetic analysis of *Ubx* involved 21 amino acid sequences. *OfUbx* was denoted by a square.

**Figure 2 insects-13-00384-f002:**
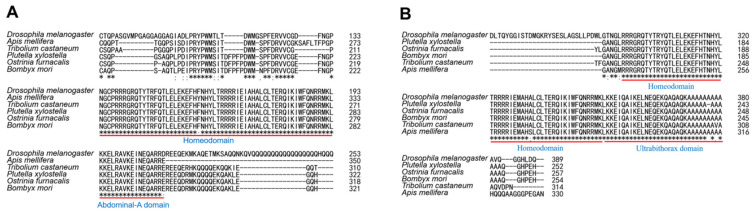
**Sequence alignment of *Abd-A* and *Ubx* proteins.** (**A**) *Abd-A* and (**B**) *Ubx* proteins were aligned from six insects, including previously reported sequences from *D. melanogaster*, *A. mellifera*, *T. castaneum*, *P. xylostella*, *B. mori*, and our newly identified sequences from *O. furnacalis*. Alignments of *Abd-A* and *Ubx* have common Homeodomain and specific conserved Abdominal-A and Ultrabithorax domains.

**Figure 3 insects-13-00384-f003:**
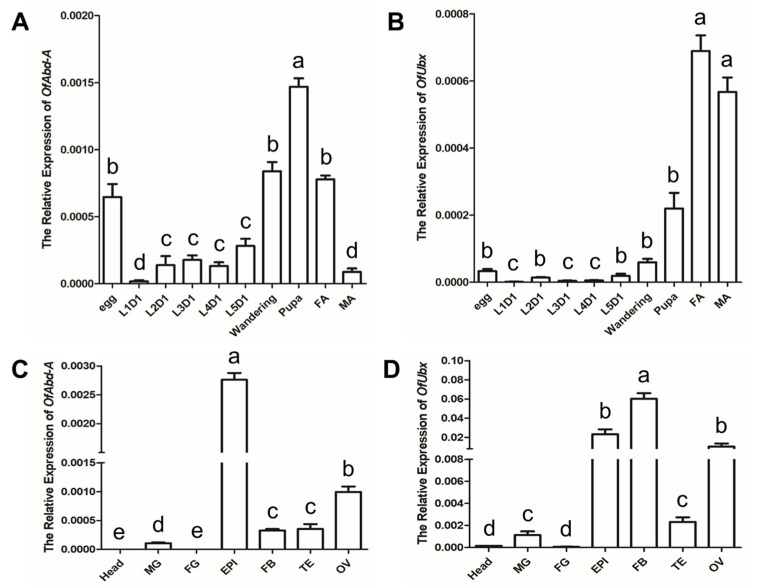
**Spatial and temporal expression patterns of *OfAbd-A* and *Ubx*.** The relative expressions of (**A**) *OfAbd-A* and (**B**) *OfUbx* in eggs, from the first day of first larval instar (L1D1) to the first day of fifth larval instar (L5D1), and at wandering stage, pupal stage, female adult stage (FA), and male adult stage (MA). The relative expressions of (**C**) *OfAbd-A* and (**D**) *OfUbx* in head, midgut (MG), foregut (FG), fat body (FB), epidermis (EPI), testes (TE), and ovaries (OV) at the third day of fifth instar (L5D3). Means labeled with different letters indicate significant difference at *p* < 0.05 (*n* = 3).

**Figure 4 insects-13-00384-f004:**
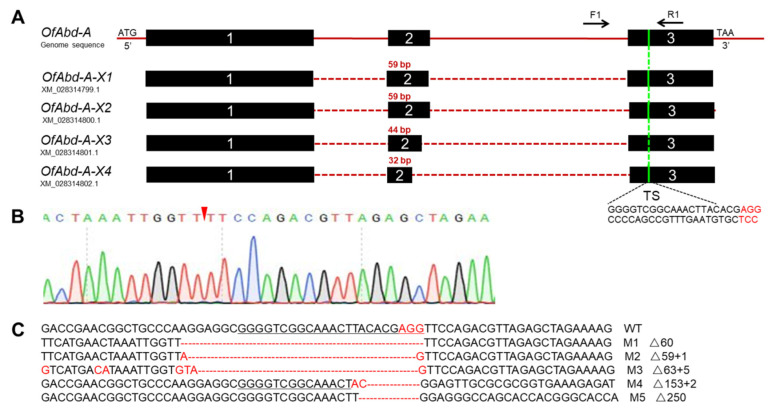
**CRISPR/Cas9-mediated mutations****within *OfAbd-A* target site****s.** (**A**) Target site of *OfAbd-A* genome locus focused on the third exon of the common region in the four splice variants. (**B**) Sequencing chromatogram of *OfAbd-A* mutants. The red wedge indicates position of cleavage by the CRISPR/Cas9 genome editing system. M1 (Mutant 1); M5 (Mutant 5). (**C**) Mutations detected by sequencing. The PAM sequence is in red. The black line represents the target site.

**Figure 5 insects-13-00384-f005:**
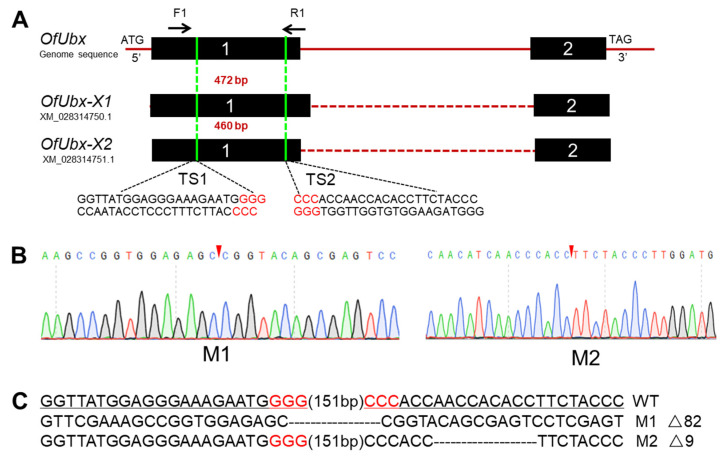
**CRISPR/Cas9-mediated mutations within *OfUbx* target sites.** (**A**) The target site of *OfUbx* genome locus in exon 1 of the common region for the two splice variants. (**B**) Sequencing chromatogram of *OfUbx* mutants. The red wedge indicates the position of cleavage by the CRISPR/Cas9 genome editing system. M1 (Mutant1); M2 (Mutant2). (**C**) Genotype detection of *OfUbx* genome sequence. The PAM sequence is in red. The black line represents the target site. The two target sites are separated by 151 bp.

**Figure 6 insects-13-00384-f006:**
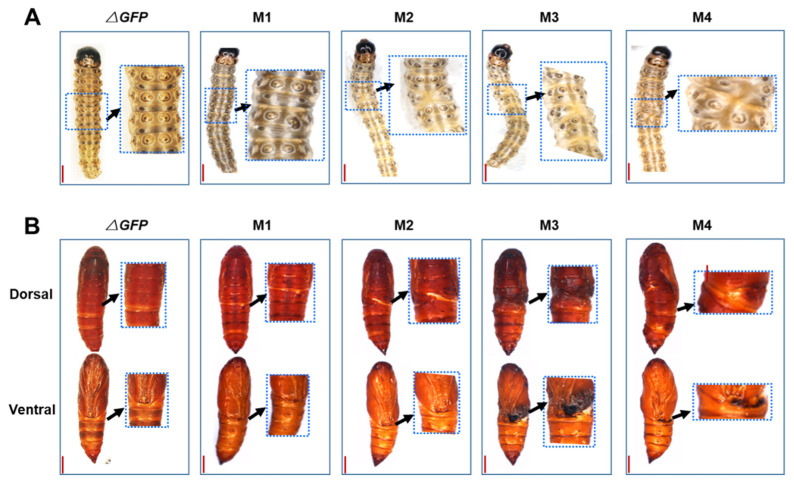
**Segmentation malformation in *OfAbd-A* mutants****at larval and pupal stages.** (**A**) The phenotypes of *OfAbd-A* mutants in the larval stage. Arrows show abnormal segments. Bar = 0.3 mm. (**B**) The phenotypes of *OfAbd-A* mutants in the pupal stage. Arrows show abnormal segments. Bar = 0.2 mm.

**Figure 7 insects-13-00384-f007:**
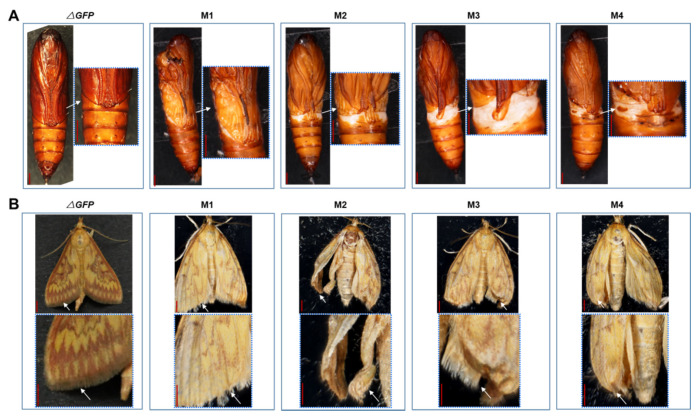
**Wing****deformation in *OfUbx* mutants****at pupal and adult stages.** (**A**) The phenotypes of *OfUbx* mutants in the larval stage. Arrows show abnormal and deficient wing discs. Bar = 0.3 mm. (**B**) The phenotypes of *OfUbx* mutants in the adult stage. Arrows show folded and deficient wings. Bar = 0.2 mm.

**Figure 8 insects-13-00384-f008:**
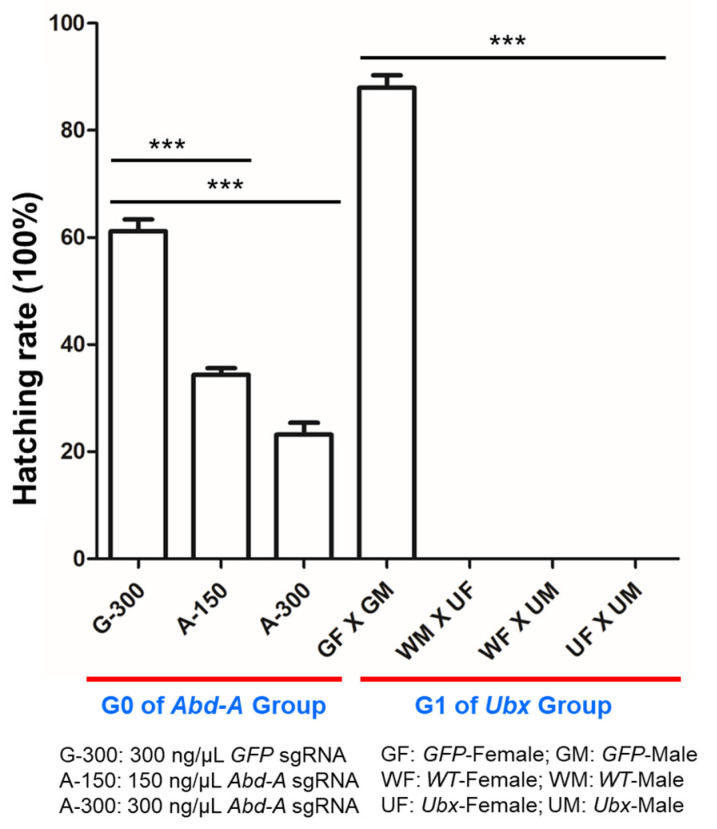
**Phenotypic impacts of *OfAbd-A* and *OfUbx* mutagenesis on *O. furnacalis* hatching rate.** Knocking out of *OfAbd-A* resulted in embryonic lethality, while *OfUbx* mutation led to adult sterility. From left to right, bars depict the hatching rates of eggs injected with different concentrations of *OfAbd-A* sgRNA, and the hatching rates of eggs produced by adult *OfUbx* mutants mated with each other. The asterisks (***) indicate significant differences (*p* < 0.01 or *p* < 0.001), compared to results from wild-type adults with a two-tailed *t*-test.

**Table 1 insects-13-00384-t001:** Mutation frequency of CRISPR/Cas9-mediated mutagenesis in *O. furnacalis*.

sgRNA	Conc (ng/μL)	Injected ^1^	Hatched ^2^	L Mutant ^3^	Pupation ^3^	P Mutant ^4^	Adult ^5^	A Mutant ^6^
*OfAbd-A*	300	785	165 (21.0)	92 (55.8)	87 (52.7)	29 (33.3)	-	-
	150	596	197 (33.1)	75 (38.1)	128 (65.1)	18 (24.0)	-	-
*Ubx*	300	486	179 (36.8)	-	118 (65.9)	53 (44.9)	62 (52.5)	18 (29.3)
	150	568	234 (41.2)	-	160 (68.4)	58 (36.2)	95 (59.4)	24 (25.3)
GFP	300	245	152 (62.0)	-	108 (71.1)	-	70 (64.8)	-

^1^ Number of injected individuals; ^2^ number and percent (%) of hatched individuals; ^3^ number and percent (%) of larvae that entered pupal stage; ^4^ number and percent (%) of pupal mutants; ^5^ number and percent (%) of pupae that entered adult stage; ^6^ number and percent (%) of adult mutants.

## Data Availability

The data presented in this study are available within the article and Appendix A.

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
