# Peer review of "CRISPR/Cas9-Mediated Mutagenesis of Abdominal-A and Ultrabithorax in the Asian Corn Borer, Ostrinia furnacalis"

_insects, 2022, doi:10.3390/insects13040384_

Round 1

Reviewer 1 Report

In the manuscript by Bi et al, the study itself is solid, but the organization of the manuscript can be improved in a few places:

Methods of spatial and temporal expression should precede mutagenesis experiments to be consistent with the ordering in results. Also, main results from this section (spatial and temporal expression) should be presented as main figures rather than supplemental figures. 

In discussion, many sections either belong to results or are redundant with results. For example, 4.1 (Phylogenetic analysis of OfAbd-A and OfUbx) should all belong to results, and there’s no discussion content whatsoever. Also, main figures ought to be added here. All main results (with their own subsections) should have at least one main figure.

Meanwhile, use of subsections can be removed altogether to form a coherent discussion. For the least, the number of subsections should be reduced. All results do not need to be referred to in discussion.

Reviewer 2 Report

The manuscript by Bi et al describes the phenotypic impacts of CRISPR-mediated knockout of two homeotic genes, Abd-A and Ubx, in the Asian corn borer Ostrinia furnicalis. The authors provide evidence of successful knockouts of the two genes, at both the DNA and RNA transcript level. They provide visual evidence of morphological defects and measurable physiological defects in the knockout mutants. The mutations observed agree with current models of the roles of these genes in insect development, and the single gene knockouts have predictable outcomes on the expression of some other homeotic genes. They finally speculate on the potential to use genetic editing in a transgenic application to control this serious pest insect. Overall, the manuscript was generally informative, most methods were adequately described, and the majority of results clearly presented. The Discussion suitably highlighted key findings and offered ideas on how the study could prove useful in pest insect control technologies. I think the manuscript will be of considerable interest to many researchers, not just interested in corn borers, but in the development of CRISPR technologies.

I have only a few suggestions for the authors to consider:

  1. The phylogenetic analyses should be in the Results section, not left until the Discussion.
  2. For Figure 1, how many larvae were examined to identify the mutant sequences? Were any of the unhatched embryos examined to see if there were more extensive deletions associated with failure to hatch?
  3. For the mutant larvae analyses of mutated sequences, were multiple PCR products sequenced from each individual to determine how many were homozygous/hemizygous for the mutations?
  4. The methodological details for Figure 5 were not described. How many G1 males and females were crossed? Were they single pair matings? Did the insects attempt to mate?
  5. For Figure S5, there are no statistical analyses (ANOVA) to determine which levels of expression differ from one another.
  6. The pleiotropic impacts of Add-A and Ubx were intriguing, and could potentially inform us of which genes are upstream/downstream of these two genes. However, Ubx and Abd-A knockouts produced opposite effects on Lab and Pb, two genes presumably upstream of Ubx and Abd-A. Is the sequence of Hox genes in Ostrinia not known? Do the opposite effects provide any insights into how the genes are arranged (as gene order tends to reflect gene expression within the Hox clusters), and interact with each other?
  7. The authors noted that genetic transformation has previously been achieved in Ostrinia (lines 441-442), but no reference is provided.
